# 6-Thioguanine Inhibits Herpes Simplex Virus 1 Infection of Eyes

Deyan Chen,[a] Ye Liu,[b] Fang Zhang,[c] Qiao You,[a] Wenyuan Ma,[a] Jing Wu,[a] (ID) Zhiwei Wu[d,e,f,g]

[a]Medical School of Nanjing University, Nanjing, Jiangsu, China

[b]Department of Ophthalmology, JinLing Hospital, School of Medicine, Nanjing University, Nanjing, Jiangsu, China

[c]Department of Burns and Plastic Surgery, Nanjing Drum Tower Hospital, School of Medicine, Nanjing University, Nanjing, Jiangsu, China

[d]School of Life Sciences, Ningxia University, Yinchuan, People's Republic of China

[e]Center for Public Health Research, Nanjing University Medical School, Nanjing, China

[f]State Key Laboratory of Analytical Chemistry for Life Science, Nanjing University, Nanjing, China

[g]Jiangsu Key Laboratory of Molecular Medicine, Medical School, Nanjing University, Nanjing, People's Republic of China

Deyan Chen and Ye Liu contributed equally to this article. Author order was determined both alphabetically and in order of increasing seniority.

**ABSTRACT** Herpes simplex virus 1 (HSV-1) infects eye corneal tissues leading to herpetic stromal keratitis (HSK), which is one of the leading causes of blindness. Here in our study, we found that 6-thioguanine (6-TG), a once clinically approved medication for child acute myelogenous leukemia, inhibited multiple strains of HSV-1 infection *in vitro* and *in vivo*. 6-TG is more potent than acyclovir (ACV) and ganciclovir (GCV), with the 50% inhibitory concentration ($IC_{50}$) of 6-TG at 0.104 $\mu$M with high stimulation index (SI) (SI = 6,475.48) compared to the $IC_{50}$ of ACV at 1.253 $\mu$M and the $IC_{50}$ of GCV at 1.257 $\mu$M. In addition, 6-TG at 500 $\mu$M topically applied to the eyes with HSV-1 infection significantly inhibits HSV-1 replication, alleviates virus-induced HSK pathogenesis, and improves eye conditions. More importantly, 6-TG is effective against ACV-resistant HSV-1 strains, including HSV-1/153 and HSV-1/blue. Knockdown of Rac1 with small interfering RNA (siRNA) negatively affected HSV-1 replication, suggesting that Rac1 facilitated HSV-1 replication. Following HSV-1 infection of human corneal epithelial cells (HCECs), endogenous Rac1 activity was upregulated by glutathione *S*-transferase (GST) pulldown assay. We further found that Rac1 was highly expressed in the corneal tissue of HSK patients compared to normal individuals. Mechanistic study showed that 6-TG inhibited HSV-1 replication by targeting Rac1 activity in HSV-1 infected cells, and the Rac1 is critical in the pathogenesis of HSK. Our results indicated that 6-TG is a promising therapeutic molecule for the treatment of HSK.

**IMPORTANCE** We reported the discovery of 6-TG inhibition of HSV-1 infection and its inhibitory roles in HSK both *in vitro* and *in vivo*. 6-TG was shown to possess at least 10$\times$ more potent inhibitory activity against HSV-1 than ACV and GCV and, more importantly, inhibit ACV/GCV-resistant mutant viruses. Animal model studies showed that gel-formulated 6-TG topically applied to eyes locally infected with HSV-1 could significantly inhibit HSV-1 replication, alleviate virus-induced HSK pathogenesis, and improve eye conditions. Further study showed that HSV-1 infection upregulated Rac1 expression, and knockdown of Rac1 using siRNA markedly restricted HSV-1 replication, suggesting that Rac1 is required for HSV-1 replication. In addition, we also documented that Rac1 is highly expressed in corneal tissues from HSK patients, indicating that Rac1 is associated with HSK pathogenesis. In view of the high potency of 6-TG, low cytotoxicity, targeting a distinct therapeutic target, we suggest that 6-TG is a potential candidate for development as a therapeutic agent for HSK therapy.

**KEYWORDS** Rac1, 6-thioguanine, HSK, HSV-1

Address correspondence to Zhiwei Wu, wzhw@nju.edu.cn.

**M**ore than 90% of the human population has herpes simplex virus 1 (HSV-1) infection in the developing world (1–3). HSV-1 can infect corneal tissues of the eyes, resulting in herpes keratitis, which leads to corneal scarring or loss of sight (4, 5). Herpetic stromal keratitis (HSK)-associated vision impairment is predominantly due to corneal scarring and neovascularization caused by inflammation (6). Despite years of research efforts, there is still no vaccine for HSV infection (7), and antiviral therapeutics are limited (8). At present, effective antiviral drugs include oral acyclovir (ACV), trifluridine (TFT) gel, and ganciclovir (GCV) gel (9–11). Trifluridine is usually administered as a topical gel for HSV keratitis in the United States. These therapeutics exhibit many shortcomings. (i) ACV eye ointment is used for patients with herpetic keratitis. However, patients often find that it is difficult to follow the treatment due to the required frequent application and blurred vision after application. (ii) Prolonged use of ACV drives the emergence of viral resistance, and ACV-resistant strains can lead to severe disease, including disseminated infection in immune-dysregulated individuals (12–15). (iii) Prolonged use of TFT leads to other ocular diseases. Therefore, there is an urgent need to develop antiviral therapeutics with distinct mechanisms against HSV.

Thiopurine drugs including 6-thioguanine (6-TG), 6-mercaptopurin, and azathioprine have been widely used as therapeutic agents since the approval of the Food and Drug Administration in the 1960s (16). 6-TG is an effective anticancer agent in clinical practice, especially in the first treatment of acute lymphoblastic leukemia (ALL) and other hematological malignancies, with the main mechanism of action involving conversion into thioguanine deoxynucleotides and subsequent incorporation into cellular DNA, which preferentially kills rapidly dividing cancer cells (16–18). Thiopurine drugs have been used in inflammatory bowel disease (IBD) treatment including ulcerative colitis (UC) and Crohn's disease (19). In addition, the active 6-TG metabolite 6-thioguanosine 5′-triphosphate was shown to suppress the small GTPase Rac1 (20), which is believed to be responsible for the anti-inflammatory effects of 6-TG in treating IBD (21). Therefore, 6-TG is an FDA-approved drug that was firstly used for the treatment of child acute myelogenous leukemia (22, 23) and later used for the treatment of IBD and was also considered as an immunosuppressive agent in organ transplantation (24, 25). Although 6-TG has been suggested as an alternative to the classical thiopurines azathioprine and 6-mercaptopurine for the treatment of IBD, there were concerns about its toxicity profile (21). Dosing regimens vary significantly depending on its use, from 10 mg per day for long-term treatment of inflammatory diseases to up to 3 mg/kg of body weight/day in acute lymphocytic leukemia treatments (26). While toxicity of 6-TG could be significant at higher dosages, it was anticipated that its use as an antiviral agent would be over a relatively short time period and that toxicity issues would likely be minimal.

6-TG mediates its immunosuppressive effects by interfering with Rac1 protein function (20). Rac1, a major component of the Rho family of small GTPases, plays an important role in various cellular signaling pathways to regulate gene transcription, cell proliferation, apoptosis, and motility (27). Many viruses employ the Rac1 protein to regulate their infection, including African swine fever virus (ASFV) (28), enterovirus 1 (29), enterovirus 71 (30, 31), rotavirus (29), and influenza viruses (32). Previous studies have shown that HSV-1 replication relies on Rac1-mediated pathways in keratinocytes, kidney cells, and human umbilical vein endothelial cells (33–35). Based on these studies, we wanted to know whether 6-TG could have direct effects on HSV-1 replication by targeting Rac1-mediated pathology.

In the current study, we found that 6-TG at a low concentration could significantly suppress HSV-1 replication. 6-TG possesses potent inhibitory activity against HSV-1 infection both *in vitro* and *in vivo*. Additionally, we proposed that 6-TG suppressed HSV-1 infection through downregulating Rac1 expression. Thus, our study suggests that the Rac1 protein is critical for HSK pathogenesis, and 6-TG is a promising therapeutic candidate to treat HSK.

Microbiology
Spectrum

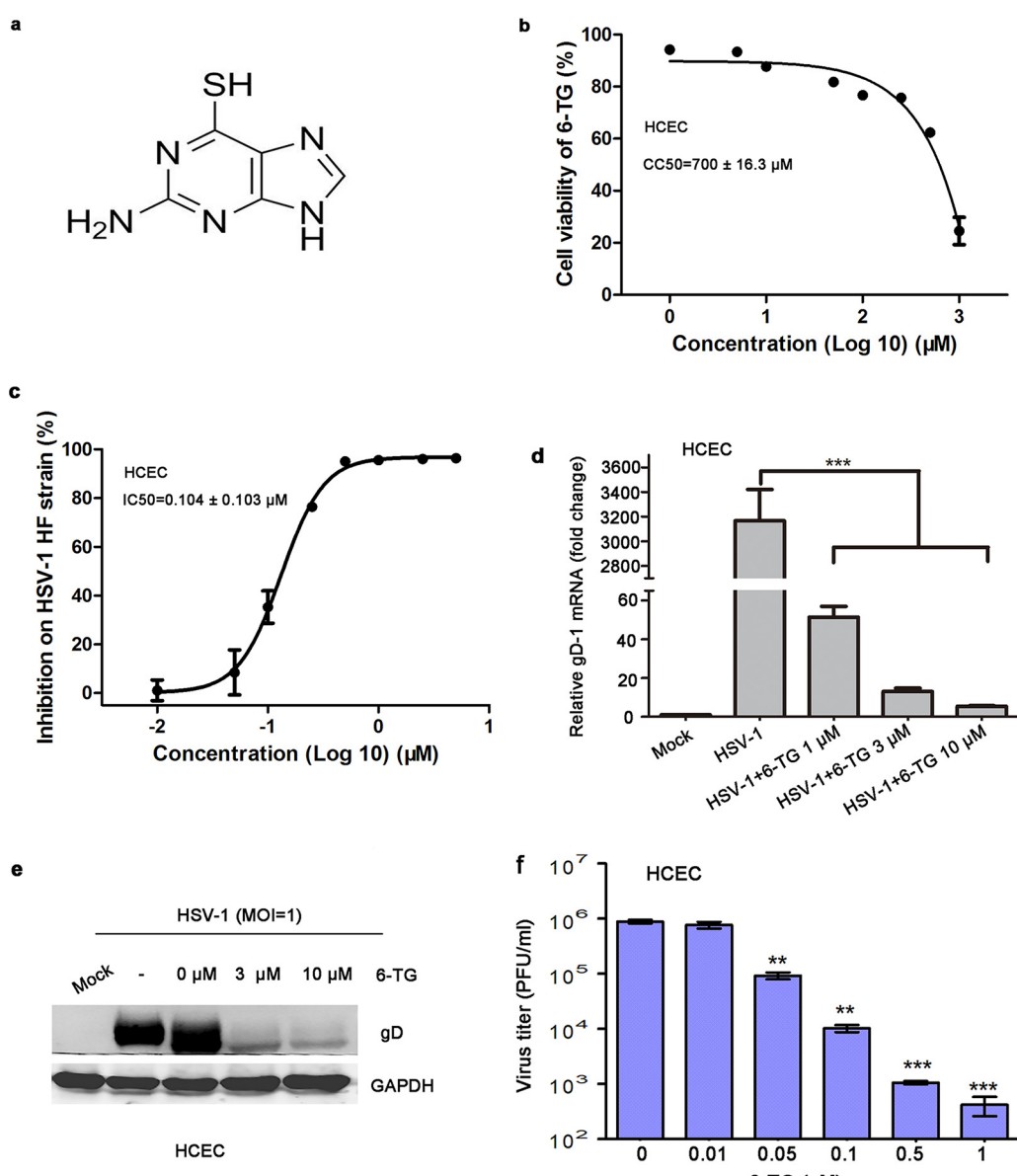

**FIG 1** 6-TG reduced HSV-1 infection of human corneal epithelial cells. (a) The structure of 6-TG is shown. (b) Viability of HCECs in the presence of 6-TG. HCECs were treated with 6-TG at different concentrations (from 0 to 1,000 $\mu$M) for 24 h and assessed by a CCK-8 kit. The data are presented as a percentage of nontreated cells. Significance between 6-TG-treated and untreated cells was determined by one-way ANOVA with Tukey's multiple-comparison test. (c) HCECs were infected with HSV-1 HF strain at an MOI of 1. After 2 h, fresh culture medium containing 6-TG (0.01, 0.05, 0.1, 0.5, 1, 5, 10, and 25 $\mu$M) was added to the cells. The data are presented as percentage of the untreated HSV-1-infected HCECs. (d) HCECs were infected with HSV-1 HF strain at an MOI of 1 in the presence of 6-TG at different concentrations (1, 3, and 10 $\mu$M) for 2 h, and then fresh culture medium containing corresponding concentrations of 6-TG was added. Mock treatment was used as a control. gD-1 mRNA of HSV-1 was determined by qPCR at 24 h. RNA was extracted from the cells 24 hours postinfection (hpi), and viral transcript gD-1 was quantified by qPCR and presented as fold change over the mock-treated cells. Significance of the change was determined by one-way analysis of variance (ANOVA) followed by Dunnett's multiple-comparison test ($n$ = 3 replicates). (e) Western blotting analysis of HSV-1 gD and GAPDH from HSV-infected HCECs in the presence of 6-TG (3 $\mu$M and 10 $\mu$M) at 24 h. HSV-1 infection was performed as described in (d). Representative blots from three independent experiments were shown. (f) Virus titers were determined by measuring PFU from virus supernatants in (c). All data are representative of three independent experiments.

## RESULTS

**6-TG suppresses HSV-1 infection of human corneal epithelial cells.** The structure of 6-TG is shown in Fig. 1a. We observed that 6-TG possessed HSV-1 inhibitory activity in an *in vitro* assay. A viral inhibitory study was performed in human corneal epithelial

**TABLE 1** 6-TG is effective against HSV-1 infection including five HSV-1 strains

| Strain | 6-TG[a] | | | ACV[a] | |
|---|---|---|---|---|---|
| | IC$_{50}$ ($\mu$M) | CC$_{50}$ ($\mu$M) | SI | IC$_{50}$ ($\mu$M) | CC$_{50}$ ($\mu$M) |
| HSV-1HF | 0.104 ± 0.013 | | 6,796.69 | 1.253 ± 0.073 | |
| HSV-1F | 0.196 ± 0.021 | | 3,571.73 | 0.209 ± 0.056 | |
| HSV-1 McKrae | 0.121 ± 0.034 | 700.06 ± 16.3 | 5,785.61 | 0.518 ± 0.047 | >5,000 |
| HSV-1/blue | 0.471 ± 0.027 | | 1,486.33 | >100 | |
| HSV-1/153 | 0.154 ± 0.019 | | 4,545.84 | >100 | |

[a]CC$_{50}$ and IC$_{50}$ of 6-TG were calculated by linear regression analysis of the viral inhibition curves in HCECs. SI was defined as the ratio of CC$_{50}$ to IC$_{50}$ (CC$_{50}$/IC$_{50}$). Data are presented as mean ± SD.

cells (HCECs) with 6-TG up to 25 $\mu$M, and the inhibition was measured by In-Cell Western blotting, quantitative PCR (qPCR), and Western blotting. We determined cytotoxicity of 6-TG on a HCEC line and observed that 20% cell death was induced by 50 $\mu$M 6-TG at 24 h, and the 50% cytotoxic concentration (CC$_{50}$) was 700 ± 16.3 $\mu$M (Fig. 1b). 6-TG inhibited HSV-1 in a dose-dependent manner with a 50% inhibitory concentration (IC$_{50}$) of 0.104 $\mu$M and a 90% inhibitory concentration (IC$_{90}$) of 1 $\mu$M (Fig. 1c), which was further supported by a 640-fold mRNA reduction at 1 $\mu$M 6-TG (Fig. 1d), a glycoprotein D (gD) reduction by 90% at 3 $\mu$M (Fig. 1e), and the significantly reduced viral particles in the culture supernatants (Fig. 1f). 6-TG was also effective against infection of five variant strains of HSV-1, including two GCV-resistant mutant strains, HSV-1/blue and HSV-1/153 (Table 1). Therefore, the biologically active concentration was well below the cytotoxicity concentration.

**6-TG is more potent than existing HSV-1 drugs.** Firstly, we tested cell viability with a standard cytotoxicity assay by using a CCK-8 kit in Vero and HeLa cells for 72 h (see Fig. S1a and b in the supplemental material). Both cells were more resistant to 6-TG than HCEC. We compared the antiviral efficacy of 6-TG with ACV and GCV in an HSV-1 HF strain infection of Vero cells. 6-TG was shown to be at least 1 log$_{10}$ more potent than both ACV and GCV (Fig. 2a). The dose-dependent analysis demonstrated that 6-TG at 0.1 $\mu$M reduced HSV-1 replication by 90% in Vero cells (Fig. 2a), in contrast to ACV and GCV at 0.1 $\mu$M that only reduced HSV-1 replication by 37.5%. 6-TG at 0.01 $\mu$M showed a 3.5 log reduction of viral particles, while ACV/GCV at 0.01 $\mu$M showed a 2 log reduction of viral particles in the culture supernatants (Fig. 2b). Western blotting analysis of gD-1 showed us that 6-TG reduced gD-1 in a dose-dependent manner, and the antiviral effect of 6-TG was better than ACV/GCV at the same concentration in Vero cells (Fig. 2c). IC$_{50}$ values for 6-TG, ACV, and GCV were 0.01 $\mu$M, 0.6 $\mu$M, and 0.5 $\mu$M, respectively, and IC$_{90}$ values for 6-TG, ACV, and GCV were 0.1 $\mu$M, 1.0 $\mu$M, and 1.0 $\mu$M, respectively. In conclusion, 6-TG potently possessed higher HSV-1 inhibitory activity than both ACV and GCV, and the concentration needed to achieve IC$_{90}$ was well below those causing cytotoxicity.

**6-TG shows a strong inhibitory activity against ACV-resistant strains.** ACV- and/or GCV-resistant mutants arise due to prolonged use of these drugs or in immunocompromised patients. We, therefore, investigated the 6-TG activity against these mutant viruses and found that 6-TG exhibited comparable inhibitory activities against both HSV-1/blue and HSV-1/153, two ACV/GCV-resistant mutant viruses. The dose-dependent analysis demonstrated that 50 $\mu$M 6-TG suppressed the replication of HSV-1/blue strain by 70% in HCECs (Fig. 3a and c), in contrast to ACV and GCV at 50 $\mu$M that did not affect HSV-1/blue strain replication. In addition, the dose-dependent analysis further demonstrated that 50 $\mu$M 6-TG suppressed HSV-1/153 replication by 80% in HCECs (Fig. 3b and d); in contrast, 50 $\mu$M ACV reduced HSV-1/153 infection by 20%, and 50 $\mu$M GCV showed a 40% reduction.

***In vitro* synergism of 6-TG and ACV against HSV-1.** As 6-TG likely acts at a distinct mechanism from that of ACV and GCV, we, therefore, tested the synergistic effect of 6-TG and ACV in blocking HSV-1 infection of Vero cells using a viral plaque assay. HSV-1 HF at an multiplicity of infection (MOI) of 0.1 was added to Vero cells in the presence of various combinations of ACV and 6-TG. At 24 h, Vero cells were fixed, and the viral

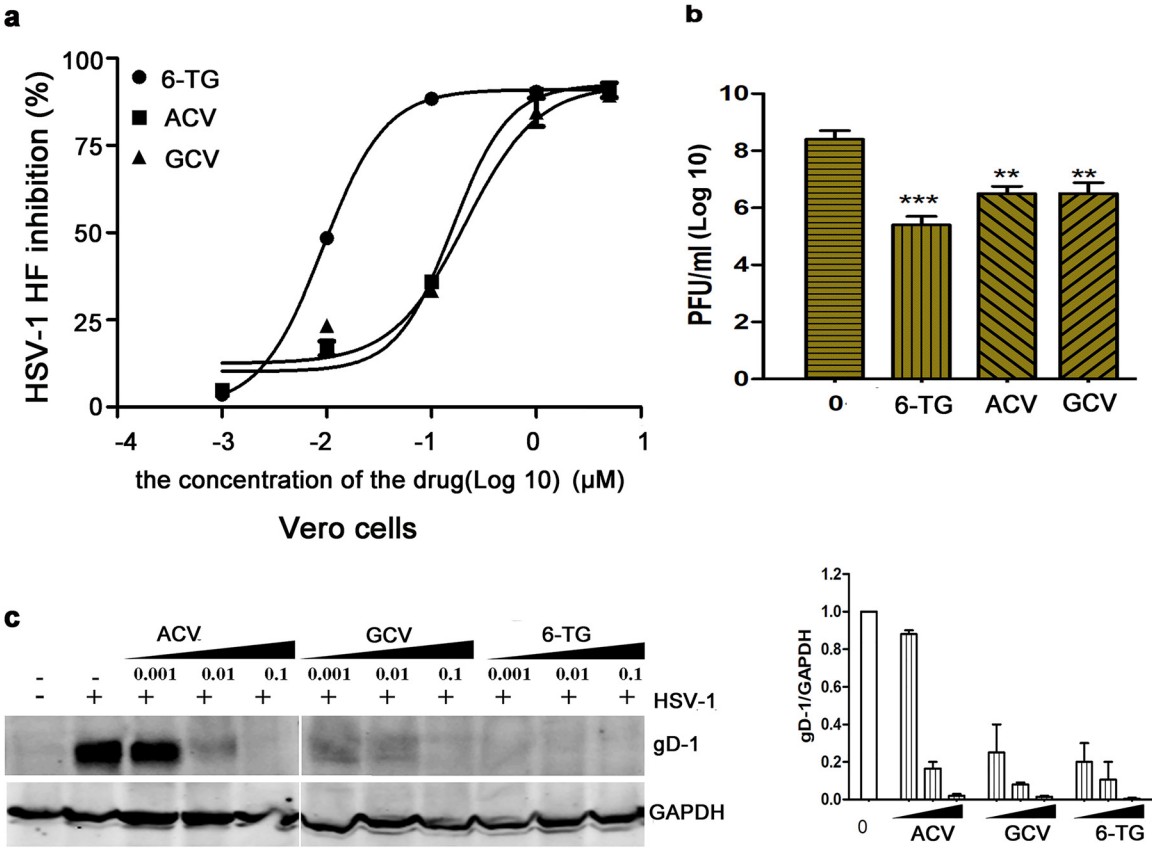

**FIG 2** 6-TG exhibited higher activity against HSV-1 than ACV and GCV. (a) The percentage of the HSV-1 HF strain infection was obtained from a viral plaque analysis. (b) Vero cells were treated with 6-TG/ACV/GCV at 0.01 $\mu$M for 2 h and then they were infected with HSV-1 HF for 24 h. Virus titers were determined by using plaque assay. The data are presented as means $\pm$ SD. One-way ANOVA was used to compare plaques in Vero cells in the presence of 6-TG, ACV, and GCV with nontreated cells. **, $P < 0.001$; ***, $P < 0.0001$. (c) Western blot presented HSV-1 gD-1 in the presence of 6-TG-treated and HSV-1-infected Vero cells. Vero cells were infected with the HSV-1 HF strain for 2 h, followed by the addition of 2% FBS of DMEM containing 6-TG/ACV/GCV (0.001, 0.01, and 0.1 $\mu$M, respectively). At 24 h, total proteins were extracted from the cells and analyzed by Western blotting assays. Representative blots from three experiments were shown. The HSV-1 gD-1 expression in the left panel relative to GAPDH was shown in the right panel ($n = 3$ replicates). One-way ANOVA with Dunnett's multiple comparison test was performed to determine the significance between the mock-treated and 6-TG/ACV/GCV-treated cells. All data are representative of three independent experiments.

protein ICP4 of HSV-1 from each drug combination was determined by In-Cell Western blotting assay. In addition, the CCK-8 assay confirmed that these combinations of 6-TG and ACV, even at the highest concentration, showed no cytotoxicity (see Fig. S2 in the supplemental material). The results were further analyzed with CompuSyn software to obtain combination index (CI) values, which would indicate if the effect of two drugs was additive, antagonistic, or synergistic. As software-determined concentrations that resulted in 50%, 75%, 90%, or 95% inhibition of HSV-1 infection, the CI values were all less than 1, indicating synergism (Fig. 4b). All of the results indicated that the combination of 6-TG and ACV worked synergistically in suppressing HSV-1 infection.

**6-TG reduces HSV-1 infection by suppressing Rac1 function.** Graber et al. and our present study published in the Biological and Pharmaceutical Bulletin showed that HSV-1 infection upregulated Rac1 activity to support its replication (35, 36). In addition, a previous study has shown that 6-TG mediated immunosuppressive effects by interfering with Rac1 protein function (20). Taken together, we wanted to know whether 6-TG reduced HSV-1 replication by suppressing Rac1. To further confirm the roles of Rac1 on HSV-1 replication, small interfering RNA (siRNA)-mediated Rac1 knockdown was used to downregulate Rac1 expression. Rac1 siRNA effectively silenced Rac1 expression in HCECs as shown by Western blot analysis (Fig. 5a). In addition, Rac1 siRNA reduced 60% of HSV-1 infection compared to the nonspecific siRNA-treated cells

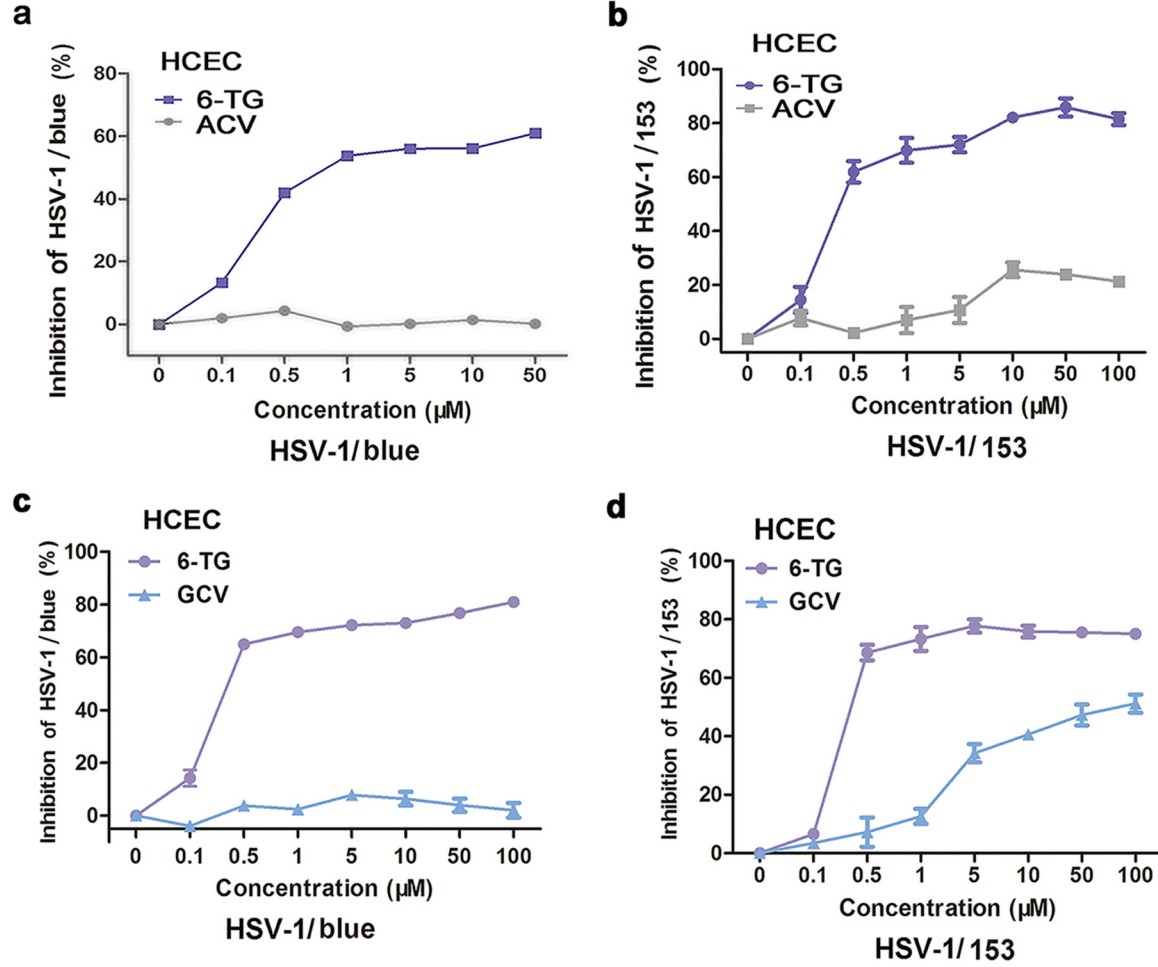

**FIG 3** 6-TG inhibited the replication of ACV/GCV-resistant HSV-1 strains. HCECs were infected with either HSV-1/blue strain (a) or HSV-1/153 strain (b) and treated with 6-TG at different concentrations (0.1, 0.5, 1, 5, 10, and 50 $\mu$M) for 24 h. HSV-1 infection was detected by In-Cell Western blotting assay. ACV treatment at the same concentration was used as a control. HCECs were infected with either HSV-1/blue strain (c) or HSV-1/153 strain (d) and then treated with 6-TG at different concentrations (0.1, 0.5, 1,5, 10, 50, and 100 $\mu$M) for 24 h. HSV-1 infection was detected by In-Cell Western blotting assay. GCV treatment at the same concentration was used as a control. All data are representative of three independent experiments.

(Fig. 5b). A Rac1-specific inhibitor NSC23766 significantly reduced HSV-1 replication in a dose-dependent manner with an IC$_{50}$ of 3 $\mu$M as shown by In-Cell Western assay, which was supported by reduced gD-1 expression as detected by Western blotting analysis (Fig. 5c). We found that HSV-1-infected HCECs at an MOI of 1 exhibited increased Rac1 activity ($\sim$3 fold) by the glutathione *S*-transferase (GST) pulldown assay (Fig. 5d). We also used the GST pulldown assay to measure Rac1 activity in 6-TG-treated HCECs and HSV-1-infected HCECs *in vitro*. Western blotting analysis was used to detect the total Rac1 protein. 6-TG-treated HCECs presented lower Rac1 activity compared to that of the untreated cells as shown in Fig. 5e (lane 2 and 3 versus lane 1). Rac1 activity increased 3-fold following HSV-1 infection compared to that of the untreated HCECs (Fig. 5f, lane 2 versus lane 1), and 6-TG treatment reduced Rac1 in HSV-1-infected cells (lane 3 and 4 versus lane 2). Therefore, we postulate that 6-TG inhibited HSV-1 infection by reducing Rac1 function.

**Rac1 is closely associated with HSK development.** HSV-1-infected corneal tissues commonly developed HSK in a mouse model. Therefore, we examined the Rac1 protein in this model and found that the Rac1 protein was upregulated in the HSV-1-infected eyes compared to that in the mock-infected animals by indirect immunofluorescence microscopy (Fig. 6a). The Rac1 protein increased 4-fold as measured by

Microbiology Spectrum

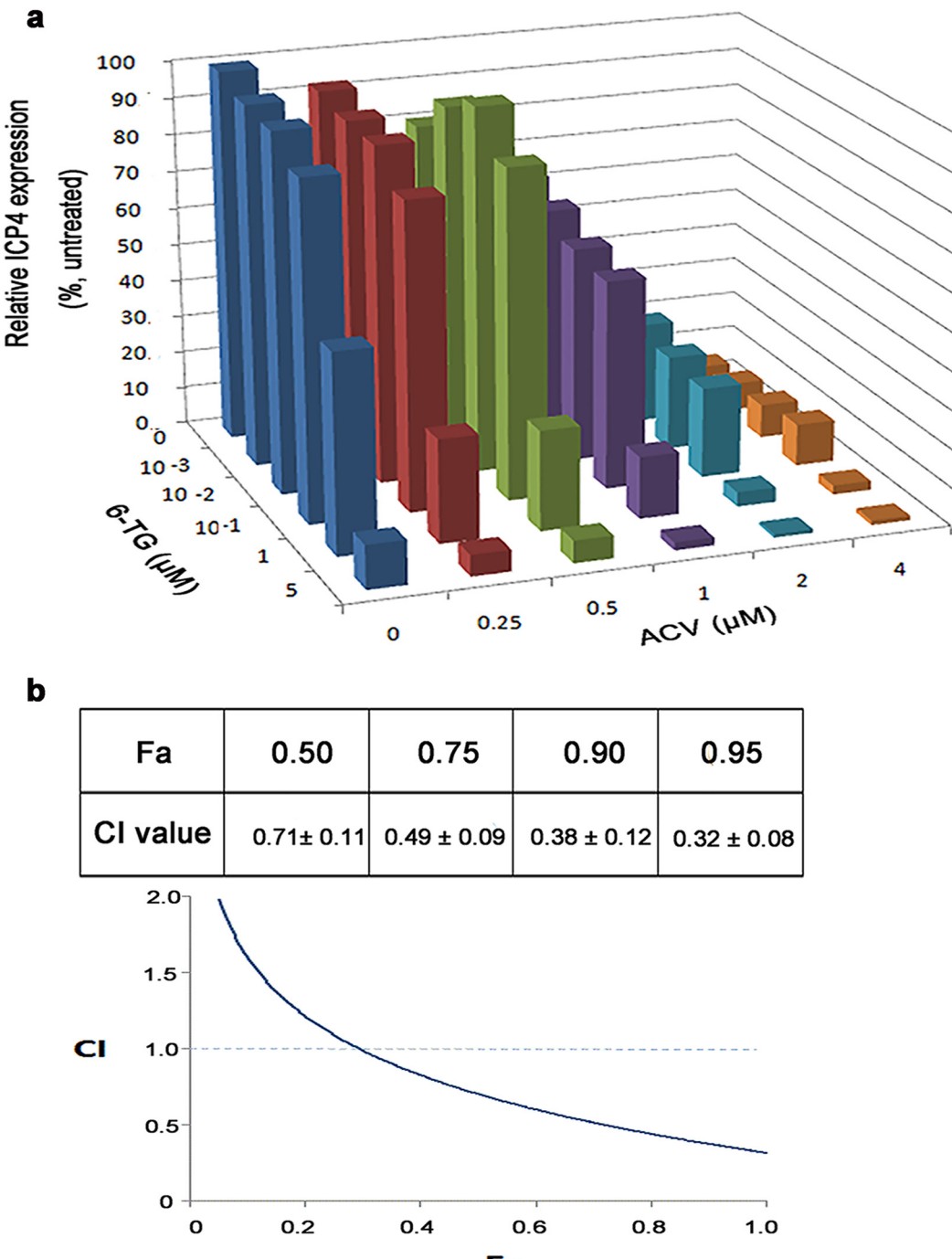

**FIG 4** 6-TG and ACV acted synergistically to suppress HSV-1 infection. (a) Three-dimensional graph depicting ICP4 protein expression at the different drug combinations. The effective concentrations for the inhibition of HSV-1 infection by a compound alone or in combination were plotted. The CI values were calculated using CalcuSyn software and are as follows: CI < 0.1, very strong synergism; CI = 0.1 to 0.3, strong synergism; CI = 0.3 to 0.7, synergism; CI = 0.7 to 0.85, moderate synergism; CI = 0.85 to 0.90, slight synergism; CI = 0.9 to 1.1, nearly additive; and CI > 1.1, antagonism. The data represent means ± SD of triplicate determinations from three independent experiments. (b) Software-determined CI values at the specified fractions affected (Fa). "Fa" refers to inhibition of HSV-1 plaque formation (fraction of control). Data are presented as means of results from three experiments ± SEM.

immunofluorescence analysis (Fig. 6b). To further confirm the role of Rac1 in the development of HSK, we firstly analyzed the Rac1 protein in the corneal tissues from five HSK patients and five normal individuals by hematoxylin and eosin (H&E) staining and immunohistochemical analysis. The corneal epithelium from HSK patients showed loss

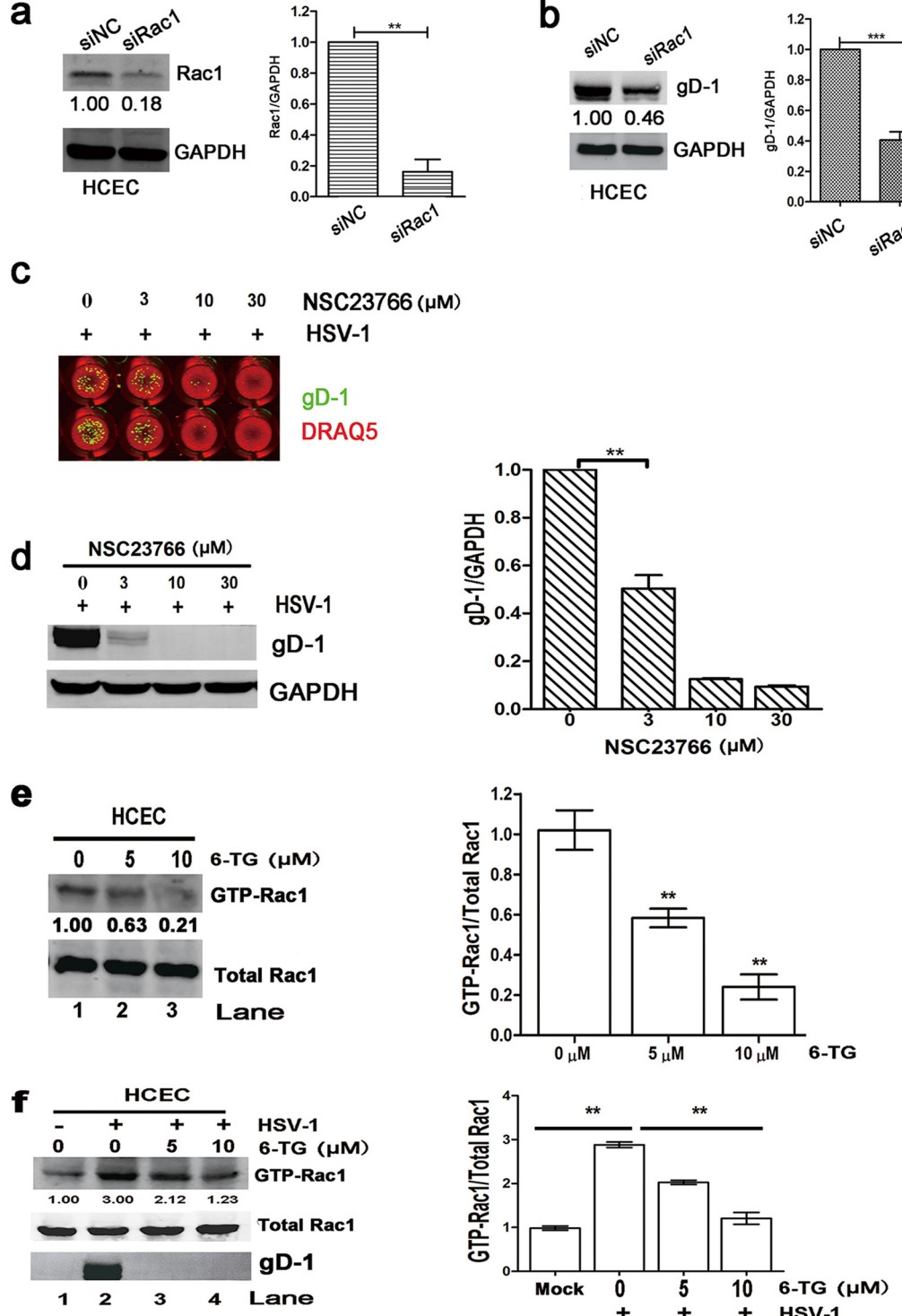

FIG 5 6-TG reduced HSV-1 replication by suppressing Rac1 activity. (a) Rac1 siRNA effectively silenced Rac1 protein in HCECs as shown by Western blotting. (b) Rac1 siRNA-treated HCECs reduced HSV-1 replication compared to that of the nonspecific siRNA-treated HCECs during HSV-1 infection. (c) NSC23766 significantly suppressed HSV-1 replication in a dose-dependent manner. In-Cell Western blotting analysis of HSV-1 gD and DRAQ5 from HSV-infected HCECs in the presence of NSC23766 (3, 10, and 30 μM) at 24 h. In addition, Western blotting analysis of HSV-1 gD and GAPDH from HSV-infected HCECs in the presence of NSC23766 (3, 10, and 30 μM) at 24 h. Significance between untreated and NSC23766-treated and HSV-1-infected cells was determined by one-way ANOVA with Tukey's multiple comparison test. (d) HCECs were infected with HSV-1 at a low MOI (MOI = 0.1) and at a high MOI (MOI = 1) for 24 h, respectively. Rac1 activity was detected by GST pulldown assay, and total Rac1 protein was measured by Western blotting assay. (e) HCECs were treated with 6-TG at 5 μM or 10 μM for 24 h, respectively. The amount of active, GTP-loaded Rac1 was

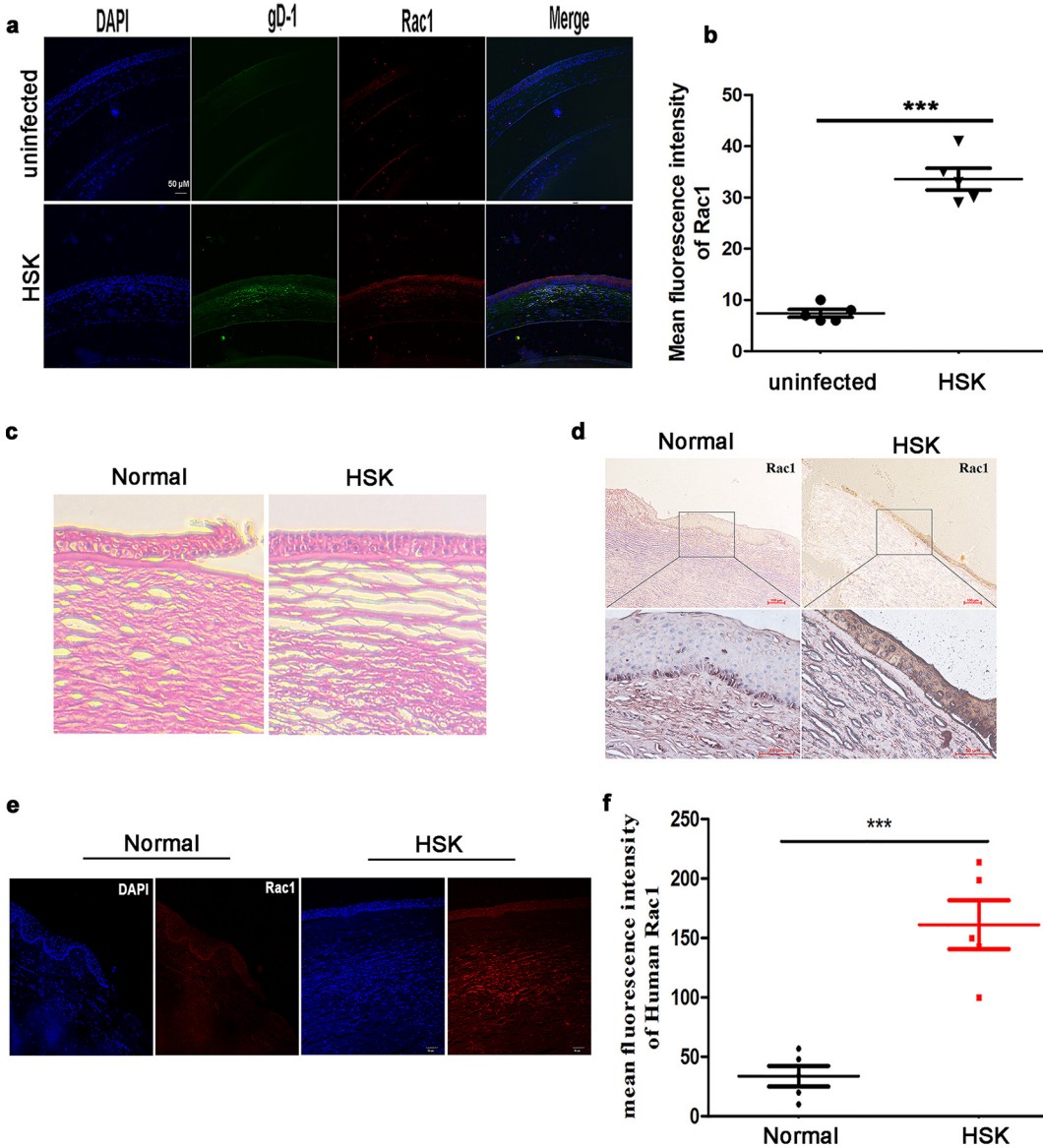

**FIG 6** Rac1 involved in the development of HSK pathogenesis. (a) Immunofluorescence staining of gD-1 (green), Rac1 protein (red), and DAPI (blue) in the corneal tissues of uninfected ($n = 5$) and HSK mice ($n = 5$), respectively. (Scale bar, 50 $\mu$m). (b) Quantification of mean fluorescence intensity of Rac1 protein in the corneal tissue from panel a is shown in the histogram ($n = 5$). (c) H&E staining of the corneal tissue of normal individuals ($n = 5$) and HSK patients ($n = 5$), respectively (scale bar, 50 $\mu$m). (d) Immunohistochemistry staining for Rac1 protein in the corneal tissues of normal individuals and HSK patients. (e) Immunofluorescence staining for Rac1 protein (red) and DAPI (blue) in the corneal tissues of normal individuals and HSK patients. (f) Quantification of mean fluorescence intensity of human Rac1 protein in the corneal tissues from panel e is shown in the histogram ($n = 5$) (Scale bar, 20 $\mu$m). $P$ values were determined by unpaired $t$ test or two-way ANOVA. Data are the means $\pm$ SD. All data are representative of three independent experiments.

of epithelial cells and became thinner than those from normal individuals (Fig. 6c). Compared to normal individuals, HSK patients had significantly increased Rac1 protein expression (Fig. 6d). Notably, immunofluorescence of the Rac1 protein increased 5-fold in the corneal tissues of HSK compared to that of normal individuals (Fig. 6e and f).

**FIG 5** Legend (Continued)

determined by using a GST pulldown assay. Western blotting analysis of Rac1 protein and gD-1 from untreated and 6-TG-treated cells. (f) HCECs were treated with 6-TG at 5 $\mu$M and 10 $\mu$M for 2 h followed by infection with HSV-1 for 24 h. The amount of active, GTP-loaded Rac1was determined by using a GST pulldown assay. Western blotting analysis of Rac1 protein and GAPDH from untreated and 6-TG-treated cells. The numbers represent the relative density of the band in comparison to the corresponding untreated control normalized to GAPDH. The value of untreated HCECs is set at 1.00 (100%). All data are representative of three independent experiments.

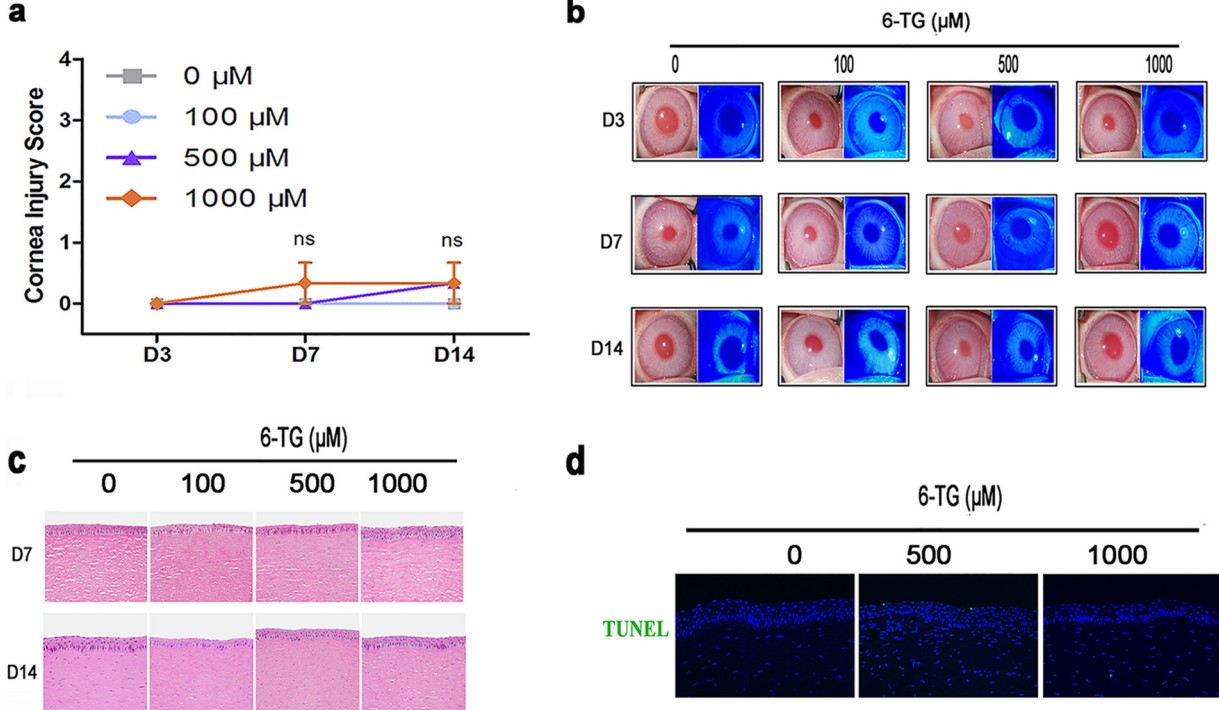

**FIG 7** 6-TG gel toxicity to cornea in a rabbit model. (a) Experimental scheme and corneal injury scores were recorded according to Table 5. Normal eyes were assigned to either no treatment or topical 6-TG gel three times daily. The corneas were examined, and the corneal injury scores were recorded on days 3, 7, and 14. Then they were collected for qPCR analysis on day 3, 7, and 14. (b) Representative corneal images on days 3, 7, and 14. Neither epithelial defect nor opacity was observed in any animals. (c) Representative images of H&E were obtained on days 7 and 14 to examine damages in corneal structure in the presence of 6-TG treatment 100, 500, and 1,000 $\mu$M). (d) Representative images of TUNEL staining were obtained on day 14 to examine the apoptosis induced by 6-TG treatment in the corneal epithelium. H&E, hematoxylin and eosin; TUNEL, terminal deoxynucleotidyltransferase-mediated nick end labeling. All data are representative of three independent experiments.

Consistently, immunohistochemical staining showed that the Rac1 protein was markedly increased in the corneal tissues of HSK but had a low basal level in normal individuals.

**Safety of topical 6-TG gel on the rabbit cornea.** Before conducting *in vivo* efficacy studies, toxicity of 6-TG gel for the corneal epithelium was evaluated. To examine the safety of topical 6-TG gel to the eyes, we applied escalating doses (100, 500, and 1,000 $\mu$M) of 6-TG gel three times daily to the ocular surface of rabbits (*n* = 3 in each concentration). The ocular surface in rabbit was treated with 6-TG gel daily without corneal injury on days 3, 7, and 14 (*n* = 3 in each group) (Fig. 7a). Corneal epithelia were assessed after fluorescein sodium staining and microscopic photographs were taken at the same magnification. The proportion of the stained area to the total corneal surface was calculated on day 3, day 7, and day 14. Rabbits without 6-TG gel treatment served as controls (*n* = 3). On day 3, day 7, and day 14, slit-lamp examination demonstrated that corneal transparency and epithelial integrity were not affected by topical 6-TG gel treatment (Fig. 7b). The rabbit corneas were collected for histological and molecular examinations on day 7 and day 14. Hematoxylin and eosin staining showed that no changes occurred in corneal structure in the presence of 6-TG treatment (100, 500, and 1,000 $\mu$M) (Fig. 7c). We further examined the apoptosis induced by 6-TG treatment on day 14 by terminal deoxynucleotidyltransferase-mediated nick end labeling (TUNEL) staining. There was no apoptosis in 6-TG-treated corneas (Fig. 7d). qPCR analysis showed that the mRNA of proinflammatory cytokines, including interleukin-1$\beta$ (IL-1$\beta$), IL-6, tumor necrosis factor alpha (TNF-$\alpha$), IL-10, and transforming growth factor $\beta$ (TGF-$\beta$), did not differ between 6-TG-treated

and the control animals on day 7 and 14 (see Fig. S3a and b in the supplemental material).

**Topical application of 6-TG gel suppresses HSK in a mouse model.** To investigate HSV-1 infection of cornea *in vivo*, we infected eyes of mice and locally and visually assessed the infected eyes and used qPCR analysis to determine HSV-1 replication in the eye cornea (see Fig. S4a in the supplemental material). The HSV-1-infected corneas showed signs of ocular infection at 3 days postinfection (dpi), and blepharitis was extremely serious by day 7 (Fig. S4b). We found ICP0 mRNA increased by 30-fold in the HSK group compared to that of the naive group (Fig. S4c). We also performed histological staining to examine the pathology of the eyes and used immunofluorescence staining for gD-1 of HSV-1 in the corneal tissues excised from normal and HSK mice. gD-1 was detected in the HSK group but not in the naive group (Fig. S4d). The HSK mice presented with damages in the corneal epithelium and epithelial thickening typically following HSV-1 infection (Fig. S4e).

6-TG inhibited HSV-1 infection with an $IC_{50}$ of 0.1 $\mu$M in HCECs. Therefore, in our pre-experiment, we tried 0.1 $\mu$M, 1 $\mu$M, 10 $\mu$M, 100 $\mu$M, and 500 $\mu$M 6-TG to treat HSK in mice (data not shown). We found that 6-TG only at 500 $\mu$M exhibited its viral inhibitory efficacy (Fig. S4f). We found that 6-TG-treated HSK mice were healthy, and blepharitis at day 7 was markedly alleviated compared with that of phosphate-buffered saline (PBS)-treated HSK mice (Fig. S4g).

The ocular gel helped to increase the residence time of the drug in the cornea and improved absorption, and we made 6-TG in gel form for ocular surface drops. Sodium hyaluronate, also known as sodium vitreous acid, is widely used in ophthalmic surgery and medication, and its safety in the eye has been demonstrated (37). The use of sodium hyaluronate gel excludes the influence of the gel medium on the therapeutic effect, so we chose sodium hyaluronate as the gel formulation. Therefore, we formulated 6-TG as an ophthalmic gel in sodium hyaluronate and evaluated the 6-TG gel in an *in vivo* mouse model. To further compare the anti-HSV-1 effect with 6-TG gel and GCV gel, the working concentration of 6-TG gel or GCV gel was 500 $\mu$M. We applied either PBS gel, GCV gel, or 6-TG gel 3 times daily for 7 days to the infected eyes and examined the outcome of the treatment through visual assessment using corneal fluorescence as depicted in the schematic presentation in Fig. 8a. Corneal fluorescein sodium staining detected with cobalt blue light showed the corneal damage of the eyes, and increased fluorescein staining corresponds to greater damage to the corneal epithelium. We observed that PBS gel-treated mice showed severe damage in the eyes, but 6-TG gel-treated mice exhibited no damage while the GCV gel-treated mice showed apparent damage to the eyes though less severe than the PBS-treated ones (Fig. 8b). Signs of ocular infection after 7 days were clearly visible in the PBS-treated eyes, but 6-TG gel- and GCV gel-treated eyes were healthy and presented no pathologies (Fig. 8c). In addition, HSV-1 infection of the mouse cornea has been shown to induce corneal neovascularization, a clinically relevant manifestation of stromal keratitis.

To determine the effect of corneal neovascularization following virus clearance from the cornea, representative images of the corneal neovascularization were obtained from the treated mice. We found that 6-TG gel-treated mice had a significant suppression of the corneal neovascularization following HSV-1 clearance compared to PBS gel-treated mice (Fig. 8d). Body weights were recorded at various time points following HSV-1 infection. Only at 6 dpi, we observed a significant loss ($P < 0.05$) of body weight in the PBS-treated mice compared to that of the 6-TG-treated mice/GCV-treated mice (Fig. 8e). The degree of herpetic epithelial keratitis was evaluated based on the criteria listed in Table 2. HSK mice treated with 6-TG and GCV had significantly ($P < 0.05$) lower degrees of HSK compared to the PBS-treated mice at 5 and 7 dpi (Fig. 8f). Corneal tissues were collected on 7 dpi, analyzed by qPCR, and the result revealed a significant reduction of gD-1 mRNA in the 6-TG gel- and GCV gel-treated groups compared to that of the PBS-treated group (Fig. 8g). HSV-1 gD-1 mRNA in the corneal tissues of 6-TG-treated eyes was not significantly different from those of the GCV-treated eyes. The PBS gel-treated mice had a significantly higher

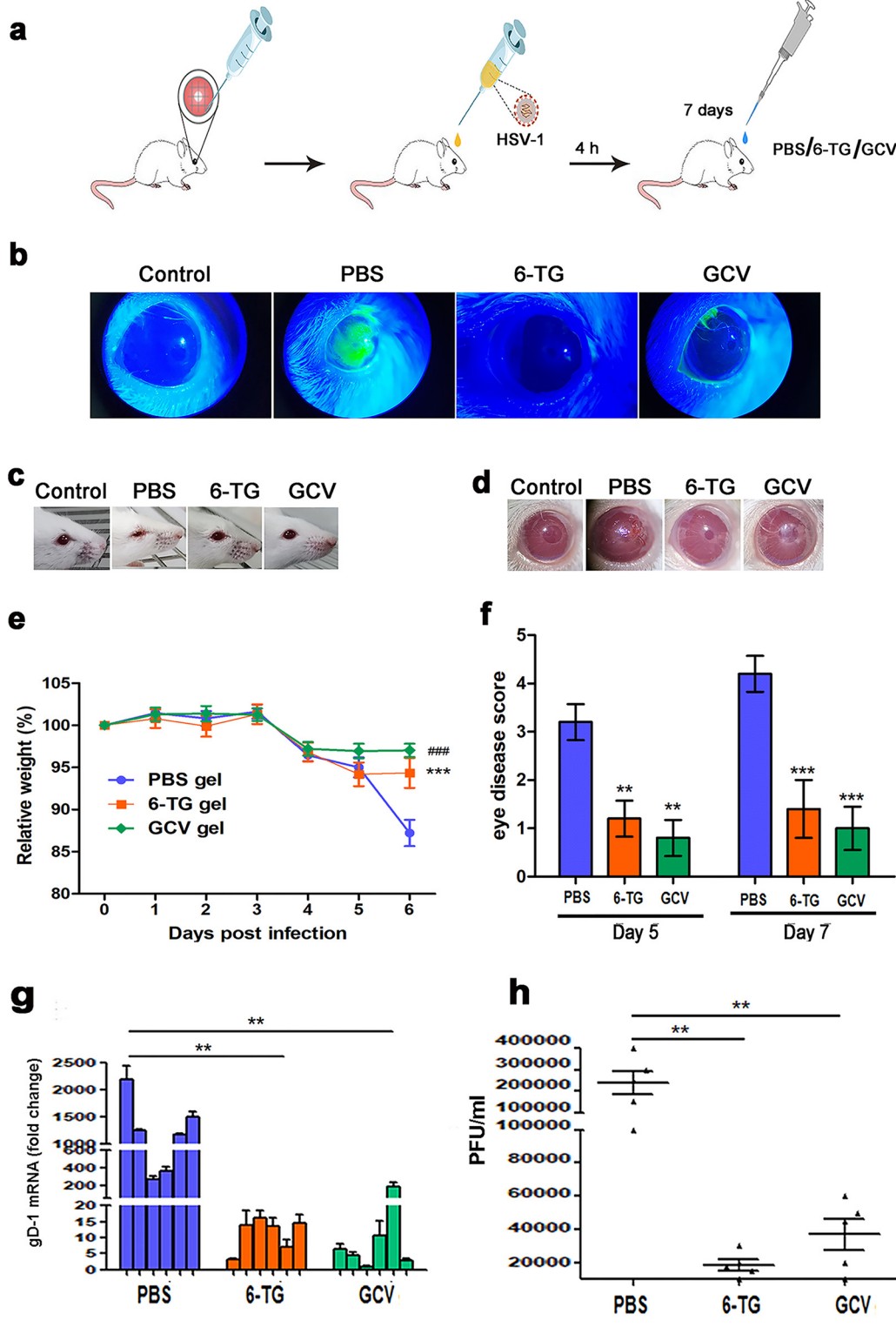

**FIG 8** Topical application of 6-TG gel reduced HSK *in vivo*. (a) Schematic of the experiment. Anesthetized BALB/c mice were placed on a holder, and the ocular surface in the corneal epithelium was scratched with a needle. The mice were then infected with $1 \times 10^6$ PFU HSV-1 HF strain in the eye cornea. After 4 h of HSV-1 infection, PBS gel, 6-TG gel (500 $\mu$M), or GCV gel (500 $\mu$M) was topically added to the eyes of HSK mice 3 times daily for 7 days. (b) Representative fluorescein images of HSK eyes treated with PBS gel, 6-TG gel, or GCV gel. HSK damage to the corneal surface can be identified by green fluorescence. Corneal fluorescein staining with cobalt blue light showed the corneal epithelium damage of the eyes. Increased fluorescein staining corresponded to greater damage in the corneal surface. Scale bar, 2 mm. (c and d) Anesthetized mice were placed on a holder and an ASX operating microscope was used to capture the images of the corneal surface of the eyes. Representative micrographs of HSK eyes treated with PBS gel, 6-TG gel, or GCV

**TABLE 2** Grading system for corneal opacity

| Grade | Description of corneal opacity |
|---|---|
| 0 | Completely transparent cornea |
| 1 | Minimal corneal opacity, but iris clearly visible |
| 2 | Moderate corneal opacity, iris vessels still visible |
| 3 | Moderate corneal opacity, pupil margin but iris vessels not visible |
| 4 | Complete corneal opacity, pupil not visible |

amount of HSV-1 in the corneal tissues, and the PBS gel-treated mice developed severe damage to the eyes and became blind in the end. At the same time, tears from the treated mice were collected and HSV-1 titers were quantified using a plaque assay. 6-TG-treated mice had a 12.5-fold reduction of PFU, and GCV-treated mice had a 6.25-fold suppression of PFU compared to that of PBS-treated mice (Fig. 8h). Together, these results indicated that topical treatment of infected eyes with 6-TG gel significantly inhibited HSV-1 replication and reduced the clinical manifestations of the infection.

## DISCUSSION

At present, therapeutic treatment of HSV infection almost entirely relies on ACV and GCV; however, prolonged use of these drugs and in immunocompromised individuals led to the emergence of drug-resistant strains (38). In addition to other clinical manifestations, HSV-1 infection of eyes leads to HSK, ultimately resulting in blindness (39, 40). The first antiviral agent approved in the United States to treat acute HSK is a 1.0% solution of TFT, a compound with tolerability issues related to its nonselective inhibition of DNA replication in both normal cells and virus-infected cells (9, 41). Another drug for treating HSK is GCV ophthalmic gel, which has selective pharmacologic activity on viral thymidine kinase (10, 11). In the current study, we showed that 6-TG is more potent than ACV and GCV, with the $IC_{50}$ of 6-TG at 0.104 $\mu$M with a high stimulation index (SI) (SI = 6,475.48) comparing the $IC_{50}$ of ACV at 1.253 $\mu$M and $IC_{50}$ of GCV at 1.257 $\mu$M, respectively. In addition, in Fig. 2a, the $IC_{50}$ of 6-TG in Vero cells seems much lower, around 0.01 $\mu$M, than in HCECs (0.1 $\mu$M). More importantly, 6-TG also inhibited two ACV/GCV-resistant strains. We further demonstrated that 6-TG and ACV worked synergistically in inhibiting HSV-1 infection in a combination of 6-TG and ACV against HSV-1.

The therapeutic target for 6-TG was previously reported to be Rac1 (20). Rac1, a major component of the Rho family of small GTPases, plays an important role in various cellular signaling pathways to regulate gene transcription, cell proliferation, apoptosis, and motility (27). The Rho-GTPase family is called the "molecular switch" because it can change between the active GTP-bound conformation and GDP-bound conformation. Many viruses employ Rac1 protein to regulate their infection. Rac1 protein was activated in the early stage of African swine fever virus (ASFV) infection and virus-induced microtubule acetylation, and intracellular transport would be impaired by suppressing Rac1 function (28). The replication of enterovirus 1 and rotavirus could be affected by Rac1 protein knockdown (29, 42). Influenza viruses were shown to be modulated by the Rac1 inhibitor through affecting the viral polymerase complex activ-

**FIG 8** Legend (Continued)

gel ($n$ = 5 per treatment group). (e) The body weight of mice was monitored for 7 days after HSV-1 infection ($n$ = 5). Data are shown as mean $\pm$ SD. ***, $P <$ 0.001, 6-TG versus HSV-1 group; ###, $P <$ 0.001, GCV versus HSV-1 group. (f) Ocular disease scores were calculated from days 5 and 7 according to the criteria in the Table 2 ($n$ = 5). (g) Secreted virus titers assessed from the swabs of right eyes. The swabs from PBS gel-, 6-TG gel-, or GCV gel-treated HSK mice were collected and used to titrate HSV-1 on Vero cells by plaques assays ($n$ = 5 per treatment group). Significance was determined by one-way ANOVA followed by Dunnett's multiple comparison test. (h) The corneal tissues were obtained from PBS gel-, 6-TG gel-, or GCV gel-treated HSK mice and were measured by qPCR to detect HSV-1 gene copies. All of the experiments were performed three times. ***, $P <$ 0.001, 6-TG versus HSV-1 group; ###, $P <$ 0.001, GCV versus HSV-1 group. Statistics were determined by one-way ANOVA followed by Dunnett's multiple comparison test. All data are representative of three independent experiments.

**TABLE 3** Details of antibodies used in this project

| Product name | Company | Product no. | Dilution (Western blotting)[a] | Dilution (immunofluorescence assay)[a] |
|---|---|---|---|---|
| Anti-ICP0 (11060) antibody | Santa Cruz | SC-53070 (lot no. A0510) | 1:500 | 1:100 |
| Anti-ICP4 (H943) antibody | Santa Cruz | SC-69809 (lot no. J1520) | 1:500 | 1:100 |
| Anti- gD 1/2 (H170) antibody | Santa Cruz | SC-69802 (lot no. L3014) | 1:500 | 1:100 |
| Anti-GAPDH antibody | ProteinTech | 60004-1-1g | 1:2,000 | N/A |
| Anti-$\beta$-actin antibody | ProteinTech | 60008-1-1g | 1:2,000 | N/A |
| Anti-rac1 antibody | Abcam | ab97732 | 1:500 | 1:100 |
| Anti-rac1 antibody | ProteinTech | 24072-1-1g | N/A | 1:100 |

[a]N/A, not applicable.

ity (32). Rac1 also served as a key regulator of actin dynamics, and Rac1 participated in the viral life cycle of HSV-1 by modulating actin cytoskeleton remodeling (43). The results from Hoppe et al. revealed that early infection of HSV-1 in MDCKII cells relied on regulated Rac1 signaling (33). The results from Petermann et al. demonstrated that HSV-1 infection induced temporary activation of Rac1, but initiation of HSV-1 infection was independent of Rac1-mediated pathway in keratinocytes (34). They both showed that increased Rac1 activity negatively impacted HSV-1 infection. Our evidence showed that Rac1 activity was upregulated upon HSV-1 infection and siRNA-Rac1 downregulation suppressed viral replication (Fig. 7), suggesting that Rac1 is required for HSV-1 replication. Our results were consistent with a previous study from Graber et al. (35). In addition, the elevated expression of Rac1 in HSK patients' corneas and the pathological thinning suggest the association of Rac1 and HSK pathogenesis. The mechanisms of Rac1 in HSV-1 replication and in HSK pathogenesis are under investigation.

The therapeutic options to treat HSK are rather limited, including trifluridine solution and GCV topical gel. More lately, therapeutic antibodies have been introduced as drugs. HSV-1 infection induced the development of corneal neovascularization through a key regulatory factor VEGF (vascular endothelial growth factor), and humanized monoclonal antibodies (MAbs) against VEGF, including bevacizumab and ranibizumab, could reduce corneal neovascularization (44–46). Previous studies also showed that T cells responses to viral antigen were the major cause of HSK (6). Rac1, as a new cellular component associated with the pathogenesis of HSK, may serve as a therapeutic target, and 6-TG targeting Rac1, with its high potency and low toxicity and its status of being an already approved drug for a different medical indication, makes it a potential candidate for development as a therapeutic agent for HSK. We further guessed that 6-TG could ameliorate HSK in the mouse model by targeting direct effects on HSV-1 replication or indirect effects on T cell-mediated pathology.

## MATERIALS AND METHODS

**Chemicals.** 6-Thioguanine, acyclovir (ACV), ganciclovir (GCV), and NSC23766 were purchased from Selleck (Houston, TX, USA), and they were dissolved in dimethyl sulfoxide (DMSO) and stored at −20℃. The final working concentrations of the drugs were prepared by diluting the stocks in fresh culture medium. Hyaluronic acid (HA) (HY-B0633) was bought from MCE company. Fluorescein sodium was obtained from the Department of Ophthalmology, JinLing Hospital, Nanjing, Jiangsu Province.

**Cell lines.** The HCEC line was purchased from Hunan Fenghui Biotechnology (Hunan, China), and the cells were grown in Dulbecco's modified Eagle's Medium (DMEM) (Thermo Fisher Scientific) and supplemented with 10% fetal bovine serum (FBS) (ExCell Bio, shanghai). The green monkey kidney cell line (Vero), human cervical carcinoma cell line (HeLa), and human immortalized keratinocytes (HaCaT) were maintained in DMEM containing 10% FBS in a 37℃ humidified atmosphere of 5% $CO_2$.

**Viruses.** HSV-1 McKrae and HSV-1 strain F were kindly provided by Kai Hu, Department of Ophthalmology, Nanjing Drum Tower Hospital, the Affiliated Hospital of the Medical School of Nanjing University, Jiangsu Province, China. HSV-1-tagged green fluorescent protein and the HSV-1 HF strain were kind gifts from Erguang Li of Nanjing University. HSV-1/blue, a TK mutant derived from HSV-1 (KOS), and one acyclovir-resistant clinical HSV-1 strain, HSV-1/153, were kindly provided by Yifei Wang, Institute of Biomedicine, College of Life Science and Technology, Jinan University, Guangzhou, China.

**Antibodies.** The information of anti-HSV-1 ICP0, anti-HSV-1 ICP4, anti-HSV-1/2 gD, and anti-glyceraldehyde-3-phosphate dehydrogenase (GAPDH) rabbit is shown in Table 3. DAPI (4′,6-diamidino-2-phenylindole) (Thermo Fisher Scientific) was used as a counter to stain the nuclei, IRDye Fluor 680-labeled IgG or IRDye Fluor 800-labeled IgG secondary antibody (LI-COR Bioscience).

**TABLE 4** Sequences of real-time PCR primer pairs used in this study

| Item | Forward (5′–3′) | Reverse (5′–3′) |
|------|------------------|------------------|
| HSV-1 gD | AGCAGGGGTTAGGGAGTTG | CCATCTTGAGAGAGGCATC |
| IL-1$\beta$ rabbit | GGTGTTGTCTGGCACGTATG | TTGGGGTCTACACTCTCCAG |
| IL-6 rabbit | CTGAAGACGACCACGATCCA | AAGGACACCCGCACTCCAT |
| IL-10 rabbit | CTCCCCTGTGAAAACAAGAG | TCCTAGACTCTAGCCGAGTT |
| TNF-$\alpha$ rabbit | ACCACGTAGCCGTGTTCAG | GCCTAGGTCTGGGTGACAAC |
| TGF-$\beta$1 rabbit | CAGCAACAATTCCTGGCGATA | AAGGCGAAAGCCCTCAATTT |
| GAPDH rabbit | GCTGAACGGGAAACTCACTG | CGAAGGTAGAGGAGTGGGTG |
| GAPDH mouse | CCATCAACGACCCCTTCATTGACC | TGGTTCACACCCATCACAAACATG |

**RNA extraction, reverse transcription, and qRT-PCR analysis.** Total RNA was extracted using TRIzol reagent (Life Technologies) and reverse transcribed using PrimeScript RT master mix for real time-PCR (RT-PCR) (TaKaRa). Quantitative real time-PCR (qRT-PCR) was performed using ABI SYBR green master mix (Life Technologies) on an ABI Prism 7500 sequence detection system. GAPDH was used for normalization of mRNA including HSV-1 viral RNA quantification. All reactions were carried out in triplicate, and analysis was carried out using the $2^{-\Delta\Delta Ct}$ method. The sequences of real-time PCR primer pairs are presented in Table 4.

**Western blotting.** Cell extracts were separated by 10% SDS-PAGE. Protein concentration was determined using a Bradford assay. After electrophoresis, proteins were transferred onto a polyvinylidene difluoride (PVDF) membrane (Millipore). The membranes were blocked for 1 h at room temperature in 2% bovine serum albumin (BSA) and then probed with primary antibodies at an appropriate dilution overnight at 4°C. The membranes were then incubated with secondary antibodies. Protein bands were visualized using an Odyssey Imaging System (LI-COR).

**An antiviral assay *in vitro*.** HCECs were treated with different concentrations of 6-TG for 2 h before HSV-1 infection. After HSV-1 infection and then being washed with DMEM, HCECs were treated with 6-TG again. HSV-1 infection was measured by In-Cell Western blot assay, and viral titers were titrated by PFU assay at 24 h postinfection after three freeze-thaw cycles. The 50% effective concentration ($IC_{50}$) of 6-TG was calculated by linear regression of the viral inhibition curves. Viral inhibition (%) was calculated as follows: viral inhibition (%) = [1 − (number of plaques) 6-TG/(number of plaques) control] × 100.

After infection with HSV-1/green fluorescent protein (GFP), Vero cells in 96 wells were incubated with serial dilution of 6-TG, and the plates were scanned using BioStack microplate stacker (BioTek Instruments, Winooski, VT, USA) at 24 h postinfection. Viral infection (%) was calculated as follows: viral infection (%) = (fluorescence$_{6\text{-TG}}$ − fluorescence$_{\text{cell control}}$)/(fluorescence$_{\text{virus}}$ − fluorescence$_{\text{cell control}}$) × 100.

**Drug synergism.** Vero cells were infected with HSV-1 HF strain at an MOI of 1. Anti-HSV-1 activity of 6-TG ($10^{-3}$, $10^{-2}$, $10^{-1}$, 1, and 5 $\mu$M) and ACV (0.25, 0.5, 1, 2, and 4 $\mu$M) were tested individually in serial concentrations in Vero cells through an In-Cell Western assay, and the 50% maximal effective concentrations ($EC_{50}$s) of the single drugs were calculated. Nontreated cells and uninfected cells were used as controls. The two drug combinations were tested at a fixed concentration ratio, which was optimized to give the greatest synergism over a range of serial dilutions. The $EC_{50}$s of single drugs and the combination index (CI) of the two drugs were calculated using CalcuSyn software according to the method of Chou and Talalay (47). The synergy was estimated from the CI values and scored as follows: CI < 0.1, very strong synergism; CI = 0.1 to 0.3, strong synergism; CI = 0.3 to 0.7, synergism; CI = 0.7 to 0.85, moderate synergism; CI = 0.85 to 0.90, slight synergism; CI = 0.9 to 1.1, nearly additive synergism; and CI > 1.1, antagonism.

**Histology, immunohistochemistry, and immunofluorescence staining.** Formalin-fixed, paraffin-embedded tissue sections (5 mm in thickness) mounted on glass slides were used for various staining. The H&E staining was performed as previously described (48). For immunofluorescence, the mouse corneal tissue sections were cryosectioned, blocked, and then stained with goat anti-rabbit Rac1 antibody (lot number 24072-1; ProteinTech), and HSK patient corneal tissue sections were stained with goat anti-rabbit Rac1 antibody (lot number 24072-1; ProteinTech). For immunochemistry, the mouse corneal tissue sections were stained with goat anti-rabbit Rac1 antibody (lot number 24072-1; ProteinTech) and anti-mouse gD-1 (lot number L3014; Santa). The fluorescence image was taken with a Nikon microscope and analyzed by ImageJ analysis software.

**Corneal toxicity in the presence of 6-TG gel treatment in a rabbit model.** Hyaluronic acid (HA) in powder was obtained in our study. Firstly, HA was dissolved in PBS at 10 mg/ml solution and sterilized by filtration through 0.45-$\mu$m filters. Then 6-TG was dissolved in the HA gel. Before conducting *in vivo* efficacy studies, toxicity of 6-TG gel for the corneal epithelium was evaluated. The New Zealand rabbit corneas were topically treated three times daily in the presence of 6-TG gel (100 $\mu$M, 500 $\mu$M, and 1,000 $\mu$M) and observed on day 3, day 7, and day 14 for corneal opacity and epithelial defects under microscope and photographed. Corneal opacity was clinically graded as follows in Table 2. Corneal epithelial defects were assessed after fluorescein sodium staining. After staining, microscopic photographs were taken at the same magnification, and the proportion of the stained area to the total corneal surface was calculated using ImageJ software (NIH).

**Animal ethics statement.** Male BALB/c mice (6 weeks old) were housed under specific pathogen-free conditions. All animal experimental procedures were approved by the Committee on the Use of Live Animals by the Ethics Committee of Nanjing Drum Tower Hospital (Nanjing, Jiangsu Province,

**TABLE 5** Corneal epithelial lesion score and description

| Score | Corneal epithelial description |
|---|---|
| 0 | Entire epithelial lesion score |
| 1 | Diffuse punctate lesion |
| 2 | Dendritic lesion occupying less than 1/4 of the entire epithelial area |
| 3 | Severe dendritic lesion extending more than 1/4 of the entire epithelial area |
| 4 | Geographic lesion on the epithelial area |
| 5 | Eye completely swollen shut |

China). We declare their compliance with published ethics. First, HA was dissolved in PBS at 10 mg/ml solution and sterilized by filtration through 0.45-$\mu$m filters. Then 6-TG and GCV were dissolved in the HA gel, respectively. The final concentration of 6-TG and GCV gel was 500 $\mu$M. The therapeutic effect of 6-TG and GCV on HSV-1-induced corneal disease was evaluated according to the previous study with some modifications (49).

Briefly, mice were anesthetized, and the corneas were scratched and infected with HSV-1 HF strain ($1 \times 10^6$ PFU) in 10 $\mu$l PBS. After infection, mice were treated with 6-TG gel or GCV gel for 7 days. Sodium hyaluronate dissolved in PBS was treated to HSV-1-infected mice as a control. In addition to daily monitoring of the body weight, the swabs of eyes from mice were harvested at 5 and 7 days postinfection and then immediately stored at $-80°$C for determination of qPCR and virus titer analysis. The scoring of eye infection (0 to 5) was performed in a blinded fashion based on the criteria reported previously (49). The criteria are set as follows in Table 5. For virus titer measurement, eye swabs were collected from the tears at 5 and 7 dpi and were titrated by PFU assay. For histopathology analysis, the right eyes of mice were removed at 7 dpi and then embedded, fixed, and sectioned. The sections were stained with hematoxylin and eosin and imaged at 20$\times$ objective using the microscope (PreciPoint M8, Freising, Germany). The thickness of the corneal epithelium was measured using the ViewPoint Software.

**Clinical human samples collection.** This study was approved by the Ethics Committee of Jiling Hospital, Medical School of Nanjing University, Nanjing, China. Every patient signed informed consent before enrollment. Corneal tissue samples from normal individuals and HSK patients were collected from April 2018 to April 2019 at the Department of Ophthalmology, Jinling Hospital, Medical School of Nanjing University, Nanjing, China. Etiologic diagnosis of the patients was confirmed by detection of HSV-1 using qPCR analysis.

**Statistical analysis.** Statistical analysis was carried out using GraphPad Prisms 6.0 software. Results were presented as means $\pm$ standard deviation (SD) with at least three biological replicates. The Student's $t$ test analysis was executed to compare the means of two groups, or one-way analysis of variance (ANOVA) was used for more than two groups. Statistical significance is as follows: *, $P < 0.05$; **, $P < 0.01$; ***, $P < 0.001$.

## SUPPLEMENTAL MATERIAL

Supplemental material is available online only.

**SUPPLEMENTAL FILE 1**, PDF file, 0.4 MB.

## ACKNOWLEDGMENTS

We thank Kai Hu in the Department of Ophthalmology, Nanjing Drum Tower Hospital, the Affiliated Hospital of the Medical School of Nanjing University, for providing HSV-1 McKrae and HSV-1 strain F. We also thank Yifei Wang in the Institute of Biomedicine, College of Life Science and Technology, Jinan University, Guangzhou, China, for providing HSV-1/blue, a TK mutant derived from HSV-1(KOS) and one acyclovir-resistant clinical HSV-1 strain HSV-1/153.

This work was supported by grants from the National Natural Science Foundation of China (81900823 to D.C. and 31970149 to Z.W.), the Major Research and Development Project (2018ZX10301406 to Z.W.), and Nanjing University-Ningxia University Collaborative Project (2017BN04 to Z.W.).

Deyan Chen and Zhiwei Wu designed the project. Deyan Chen and Ye Liu performed most of the experiments *in vitro*. Deyan Chen, Ye Liu, and Zhiwei Wu analyzed the data. Ye Liu and Fang Zhang performed the animal experiments and provided technical assistance. Wenyuan Ma and Qiao You helped to do the cytotoxicity of the rabbit experiment at the end of the project. Jing Wu provided professional statistical analysis in the data analysis. Deyan Chen and Zhiwei Wu prepared the manuscript. Zhiwei Wu monitored and financially supported the study and revised the manuscript. All authors reviewed the manuscript and approved the final version.

We declare no conflicts of interest.

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
