## [Reviewer comments · Microbiology Spectrum]

**Microbiology
Spectrum**

6-Thioguanine inhibits herpes simplex virus 1 infection of eyes

Deyan Chen, Ye Liu, Fang Zhang, Qiao You, Wenyuan Ma, Jing Wu, and Zhiwei Wu

Corresponding Author(s): Zhiwei Wu, Center for Public Health Research

Review Timeline:

Submission Date:	July 23, 2021
Editorial Decision:	August 21, 2021
Revision Received:	August 22, 2021
Accepted:	September 10, 2021

Editor: Clinton Jones

Reviewer(s): The reviewers have opted to remain anonymous.

Transaction Report:

DOI: <https://doi.org/10.1128/Spectrum.00646-21>

August 21, 2021

Dr. Zhiwei Wu
Center for Public Health Research
Medical School
22# Hankou Road
Nanjing, Jiangsu 210093
China

Re: Spectrum00646-21 (6-Thioguanine inhibits herpes simplex virus 1 infection of eyes)

Dear Dr. Zhiwei Wu:

Thank you for submitting your manuscript to Microbiology Spectrum. When submitting the revised version of your paper, please provide (1) point-by-point responses to the issues raised by the reviewers as file type "Response to Reviewers," not in your cover letter, and (2) a PDF file that indicates the changes from the original submission (by highlighting or underlining the changes) as file type "Marked Up Manuscript - For Review Only". Please use this link to submit your revised manuscript - we strongly recommend that you submit your paper within the next 60 days or reach out to me. Detailed information on submitting your revised paper are below.

Link Not Available

Sincerely,

Clinton Jones

Journals Department
Reviewer comments:

Reviewer #2 (Comments for the Author):

The purpose of this project was to evaluate 6-TG for its anti-herpes simplex virus activity and mechanism of action. The authors present data on the potency and cytotoxicity of 6-TG in several cell types. 6-TG was potent and synergized with ACV, indicating that it likely acted through a different pathway. The effects of HSV-1 and 6-TG on the Rac1 protein were investigated in cells and infected corneal tissue. 6-TG was shown to be tolerated in rabbit eyes at high concentrations, so mouse studies were undertaken. 6-TG gel prevented HSK as well as GCV gel in mouse eyes. This project includes many aspects of antiviral evaluation. It is clearly written, although the Abstract needs more details of the results and less background information.

The Abstract contains information that belongs in the Introduction (what drugs are used in which countries, drug resistance) and lacks information from the Results (IC₅₀ of 6-TG, effective dose in mice). It is not helpful to state that 6-TG is "better" than ACV and GCV because it is more potent. If 6-TG is potent but not absorbed well, then it is a "worse" drug. Stating that 6-TG is more potent than ACV and GCV is the better term. However, 6-TG is only slightly more potent than ACV (EC₅₀ approximately 0.5 micromolar), so this sentence should probably be moved to the Discussion.

Line 72-73: TFT is not given orally for HSV keratitis. Trifluridine is usually administered as a topical gel (Viroptic) for HSV keratitis. It is administered orally as a combination drug trifluridine/tipiracil for cancer treatment.

Line 109-110: Correct the grammar in this sentence. Consider this change: 6-TG mediates its immunosuppressive effects by interfering with Rac1 protein function (20).

Figure 2: In Fig. 2a, the EC₅₀ of 6-TG in Vero cells seems much lower, around 0.01 micromolar than in HCEC cells (0.1 micromolar). This point should be raised in the discussion. In Fig. 2b, the y-axis label and the figure legend do not describe the units of the plaque assay. Does the axis show the Log₁₀ of the pfu/mL? What are the error bars? How many replicates? In Fig. 2d, the scale bar on the fluorescence micrographs is too small to see and there is no DNA stain to indicate how many cells are in each field. Fig. 2d micrographs and bar graph are not necessary and should be removed.

Fig. 5e: the concentrations of 6-TG on the western blot, (0 5 10) need a label on the right side (6-TG μ M) for clarity. It would also help if it was indicated on the graph that these cells were not infected with HSV-1.

Fig. 7: The cytokine analysis is difficult to interpret because the labels on the graphs are too small. Since the data are negative, meaning 6-TG did not induce inflammation, these graphs can be moved to the Supplemental data section and enlarged enough to see the axes. Alternatively, the entire figure can be moved to the Supplement.

Fig. 8: Make the colors for each group of mice the same. Why are the colors in 8e different than 8f and 8g? Use one color for PBS, another color for 6-TG, and another color for GCV. Make all graphs consistent.

Staff Comments:

Preparing Revision Guidelines

To submit your modified manuscript, log onto the eJP submission site at <https://spectrum.msubmit.net/cgi-bin/main.plex>. Go to Author Tasks and click the appropriate manuscript title to begin the revision process. The information that you entered when you first

submitted the paper will be displayed. Please update the information as necessary. Here are a few examples of required updates that authors must address:

For complete guidelines on revision requirements, please see the journal Submission and Review Process requirements at <https://journals.asm.org/journal/Spectrum/submission-review-process>.

Submissions of a paper that does not conform to Microbiology Spectrum guidelines will delay acceptance of your manuscript. "

Please return the manuscript within 60 days; if you cannot complete the modification within this time period, please contact me. If you do not wish to modify the manuscript and prefer to submit it to another journal, please notify me of your decision immediately so that the manuscript may be formally withdrawn from consideration by Microbiology Spectrum.

If you would like to submit an image for consideration as the Featured Image for an issue, please contact Spectrum staff.

September 10, 2021

Prof. Zhiwei Wu
Center for Public Health Research
Medical School
22# Hankou Road
Nanjing, Jiangsu 210093
China

Re: Spectrum00646-21R1 (6-Thioguanine inhibits herpes simplex virus 1 infection of eyes)

Dear Prof. Zhiwei Wu:

Your manuscript has been accepted, and I am forwarding it to the ASM Journals Department for publication. You will be notified when your proofs are ready to be viewed.

Sincerely,

Clinton Jones
Editor, Microbiology Spectrum
